



# Assessing the potential for ice flow piracy between Totten and Vanderford glaciers, East Antarctica

Felicity S. McCormack[1], Jason L. Roberts[2], Bernd Kulessa[3,4,5], Alan Aitken[6,7], Christine F. Dow[8,9], Lawrence Bird[1], Benjamin K. Galton-Fenzi[2,5,10], Katharina Hochmuth[5], Richard S. Jones[1], Andrew N. Mackintosh[1], and Koi McArthur[8]

[1]Securing Antarctica's Environmental Future, School of Earth, Atmosphere & Environment, Monash University, Clayton, Victoria, Australia
[2]Australian Antarctic Division, Kingston, Tasmania, Australia
[3]School of Biosciences, Geography and Physics, Swansea University, Swansea, UK
[4]School of Technology, Environments and Design, University of Tasmania, Hobart, Tasmania, Australia
[5]The Australian Centre for Excellence in Antarctic Science, University of Tasmania, nipaluna/Hobart, Tasmania, Australia
[6]School Of Earth Sciences, The University Of Western Australia
[7]The Australian Centre for Excellence in Antarctic Science, The University of Western Australia, Perth, Western Australia, Australia
[8]Department of Applied Mathematics, University of Waterloo, Waterloo, Canada
[9]Department of Geography and Environmental Management, University of Waterloo, Waterloo, Canada
[10]The Australian Antarctic Program Partnership, Institute for Marine and Antarctic Studies, University of Tasmania, nipaluna/Hobart, Tasmania, Australia

**Correspondence:** Felicity S. McCormack (felicity.mccormack@monash.edu)

**Abstract.** The largest regional drivers of current surface elevation increases in the Antarctic Ice Sheet are associated with ice flow reconfiguration in previously active ice streams, highlighting the important role of ice dynamics in mass balance calculations. Here, we investigate controls on the evolution of the flow configuration of the Vanderford and Totten Glaciers – key outlet glaciers of the Aurora Subglacial Basin, the most rapidly thinning region of the East Antarctic Ice Sheet. We

5   synthesise factors that influence the ice flow in this region, and use an ice sheet model to investigate the sensitivity of the catchment divide location to changes in surface elevation due to thinning at Vanderford Glacier associated with ongoing retreat, and thickening at Totten Glacier associated with an intensification of the east-west snowfall gradient. The present-day catchment divide between the Totten and Vanderford glaciers is not constrained by the geology or topography, but is determined by the large-scale ice sheet geometry and its long-term evolution in response to climate forcing. Furthermore,

10  the catchment divide migrates under relatively small changes in surface elevation, leading to ice flow and basal water piracy from Totten to Vanderford Glacier. Our findings show that ice flow reconfigurations do not only occur in regions of West Antarctica like the Siple Coast, but also in the east, motivating further investigations of past, and potential for future, ice flow reconfigurations around the whole Antarctic coastline. Modelling of ice flow and basal water piracy may require coupled ice sheet thermomechanical and subglacial hydrology models, constrained by field observations of subglacial conditions. Our

15  results have implications for ice sheet mass budget studies that integrate over catchments, and the validity of the zero flow assumption when selecting sites for ice core records of past climate.



## 1 Introduction

The Vanderford Glacier is the fastest retreating glacier in the East Antarctic Ice Sheet (EAIS), with approximately 18.6 km of grounding line retreat observed over the period 1996 to 2020 (Picton et al., 2022). The Vanderford Glacier drains part of the ice contained in the Aurora Subglacial Basin (ASB; Fig. 1a), which contains approximately 7 m of global sea level equivalent (Morlighem et al., 2020) of which 3.5 m is grounded below sea level. The ASB is the most rapidly thinning basin in East Antarctica (Smith et al., 2020) and modelling studies indicate that it will continue to be the dominant East Antarctic contributor to global sea level rise over the coming decades to centuries (Pelle et al., 2020, 2021; Seroussi et al., 2020). Paleoclimate evidence suggests markedly reduced ice volumes in the ASB, Wilkes Subglacial Basin and Recovery Basins during the mid-Pliocene (3.3 Ma to 3 Ma), when global temperatures were ∼2.5 to 4°C warmer than the 1850 to 1900 mean (Foster and Rohling, 2013; Fox-Kemper et al., 2021; Wilson et al., 2018), highlighting the potential of the ASB to reach a tipping point as the climate warms (McKay et al., 2022; Noble et al., 2020).

Currently, ice flow, ice discharge and sediment discharge from the Totten Glacier is larger than at the Vanderford Glacier (Hochmuth et al., 2020; Li et al., 2016; Rignot et al., 2019). However, paleo-sediment records indicate that the Vanderford Glacier has likely been the dominant contributor to offshore sedimentation along the Sabrina and Knox Coast sectors on geologic (million year) timescales, rather than the Totten Glacier (Hochmuth et al., 2020). In particular, sedimentation rates at Vanderford Glacier were twice those at Totten Glacier during the mid-late Oligocene (27 Ma to 24 Ma), when global temperatures were approximately 3 to 4°C higher than the present-day (Westerhold et al., 2020). Furthermore, modelling indicates similar levels of erosion potential for the bed under both the Totten and Vanderford Glaciers – and indeed, that these glaciers have the highest erosive potential in East Antarctica – such that similar magnitudes of sediment discharge from each of these two glaciers are possible under present-day ice sheet geometries (Aitken and Urosevic, 2021).

Changes in basal water accumulation and routing have been hypothesised to play a role in flow diversion (piracy) – and even ice flow stagnation – between neighbouring ice streams in the past. For example, Alley et al. (1994) proposes that a diversion of basal meltwater from Kamb Ice Stream into Whillans Ice Stream, associated with an inland extension of the ice streams due to increased basal melt over the Holocene, led to a reduction in basal lubrication and meltwater production beneath Kamb Ice Stream and its consequent stagnation. These changes may have also impacted the slowdown of the adjacent Whillans Ice Stream and may impact the likelihood of future stagnation here (Beem et al., 2014; Joughin and Tulaczyk, 2002). More recent modelling of this system indicates that basal hydromechanical processes, including thermal switching between basal melting and freezing states, are a dominant control on the ice flow configuration, and that further large-scale flow reconfiguration could be possible in the next tens to hundreds of years (Bougamont et al., 2015), consistent with century-scale changes in the routing of ice streams in this region (Catania et al., 2012; Hulbe and Fahnestock, 2007).

Basal water accumulation and routing has also been suggested to play a role in the stagnation of Carlson Inlet – a proposed relict ice stream – and reconfiguration of ice flow from the Carlson Inlet to the Rutford Ice Stream over 240 years ago (Vaughan et al., 2008). Changes in basal water routing are strongly linked to ice sheet geometry changes, and modelling by Vaughan et al. (2008) suggests that thickening of the Rutford Ice Stream of only 4 % would be sufficient to re-route subglacial water to Carlson



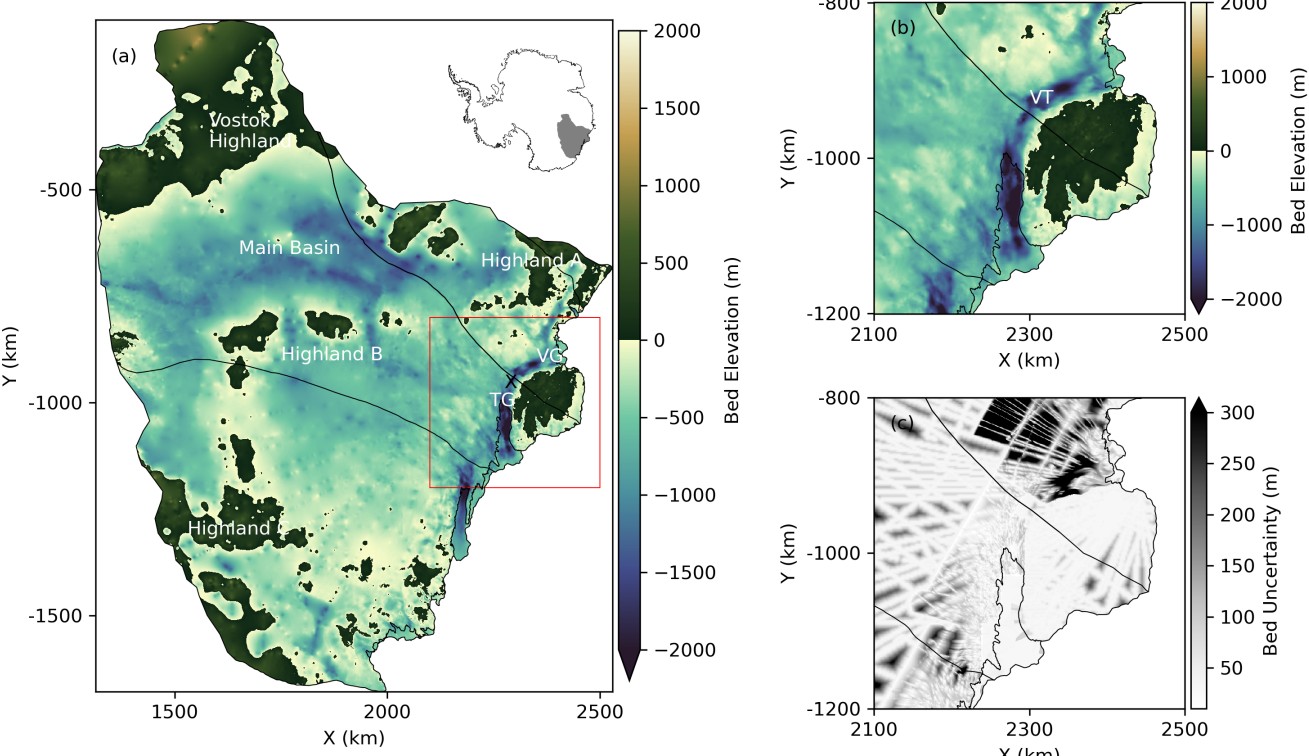

**Figure 1.** Aurora Subglacial Basin bed elevation and uncertainty derived from the BedMachine Antarctica v3 dataset (Morlighem et al., 2020). (a) Bed topography (m), with red zoom inset shown in panel (b), and (c) uncertainty (m). The ASB catchment outline is from Zwally et al. (2012) and the coastline and grounding line (black) are from MEaSUREs2. All data are referenced to the WGS84 ellipsoid and displayed in eastings (x-axis) and northings (y-axis) using polar stereographic coordinates (km). Locations of regions in panel (a) are: Vostok Highland; ASB main basin; Highland A; Highland B; Highland C; Sabrina Subglacial Basin (SSB); Totten Glacier (TG); Vanderford Glacier (VG); Sabrina Coast (SC); Knox Coast (KC); and the Elcheikh saddle point (black X). The Vanderford Trench (VT) is highlighted in panel (b).

Inlet, potentially reactivating its flow. Similar mechanisms may have been involved in the reconfiguration of ice streams in the Weddell Sea sector in the past few thousand years (Siegert et al., 2013, 2019). The precise drivers of ice flow and basal water routing changes are uncertain, and likely to vary between ice stream pairs due to variations in the basal hydromechanics, topography and ice sheet geometry, geology, and climate. Importantly, ice flow reconfiguration could be an important process
for multiple regions around the Antarctic margin, occurring even under minor changes in ice sheet geometry.

   Given the significant potential of the ASB to raise global sea levels, it is essential to understand controls on the present-day flow configuration between the Totten and Vanderford Glaciers and its potential evolution as the climate warms. In this study, we first conduct a synthesis of the geology, topography, subglacial hydrology and geomorphology of the ASB to ascertain how these factors may influence the past and present flow configuration of the Totten and Vanderofrd glaciers, and climate drivers





in the ASB and how they might change in the future. We then use an ice sheet model to generate ice surface elevation change fields to investigate the sensitivity of ice flow and basal water piracy between the Totten and Vanderford glaciers.

## 2 Controls on the ASB flow configuration

### 2.1 Current flow and trends

Totten Glacier is one of the fastest-flowing glaciers in East Antarctica (Rignot et al., 2011, 2019), with ice surface speeds
over $750\,\mathrm{m\,year^{-1}}$ at the main tributary grounding line (southernmost region) – more than $100\,\mathrm{m\,year^{-1}}$ greater than the corresponding ice surface speed at the Vanderford grounding line (Fig. 2a). Average ice discharge from 2009 to 2017 at Totten Glacier ($71.4\pm2.6\,\mathrm{Gt\,year^{-1}}$) is almost twice that at Vanderford Glacier ($36.2\pm0.5\,\mathrm{Gt\,year^{-1}}$), naturally reflecting thicker ice, a steeper surface slope, and hence greater driving stresses (Fig. 2c) near the Totten Glacier grounding line. Both glaciers have accelerated in recent decades: Vanderford Glacier surface speeds increased by 31 % over the period 2000 to 2013, with little
subsequent change (Picton et al., 2022); Totten Glacier experienced insignificant increases in surface speed from 2007 to 2022 (Rignot et al., 2022), but with quasi-decadal cyclicity (Roberts et al., 2018). Furthermore, the Vanderford Glacier grounding line retreated by ∼18.6 km over the period 1996 to 2020 (Picton et al., 2022), making it the fastest-retreating glacier in East Antarctica (Stokes et al., 2022). Changes at both glaciers are consistent with ocean-driven ice shelf thinning and calving (Sect. 3.2) and consequent reductions in buttressing (Fürst et al., 2016; Gudmundsson et al., 2019).

Discharge rates from each glacier are related to the broad-scale flow configuration of the ASB (Fig. 2). The Totten Glacier is currently fed by a catchment of 267,904 km². The majority of discharge through Totten is channelled from the west through its main tributary, with a smaller portion channelled through the eastern flank of the glacier (Fig. 2b). Ice sheet models suggest ongoing grounding line retreat along the eastern flank and into the Sabrina Subglacial Basin (SSB) will occur in coming decades (McCormack et al., 2021; Pelle et al., 2020; Sun et al., 2016). Despite the fact that the undulating bed topography
in the SSB lies largely below sea level (Fig. 1), this portion of the Totten catchment is generally thinner, having a sea level potential that is much smaller than that of the ASB main basin (Morlighem et al., 2020).

The current Vincennes Bay catchment that feeds the Vanderford Glacier is approximately 71,329 km² – just over a quarter of the size of the adjacent Totten catchment. Similar to Totten, discharge from Vanderford Glacier is channelled through a western and an eastern trunk (Fig. 2b). The western trunk largely originates from, and to the west of, Highland A (compare
Figs. 1 and 2). The eastern trunk originates in the same region as the main Totten tributary, east and south of Highland A; the Elcheikh saddle point (cross in Fig. 1a) marks the divergence of flow to either the main Totten tributary or the eastern trunk of the Vanderford Glacier.

### 2.2 Geology

The geology of the ASB – in particular, the extents and types of sedimentary basins in the Totten and Vincennes Bay catchments
– shows correspondence with patterns in ice sheet flow and subglacial hydrology (section 2.4). The ASB region is dominated





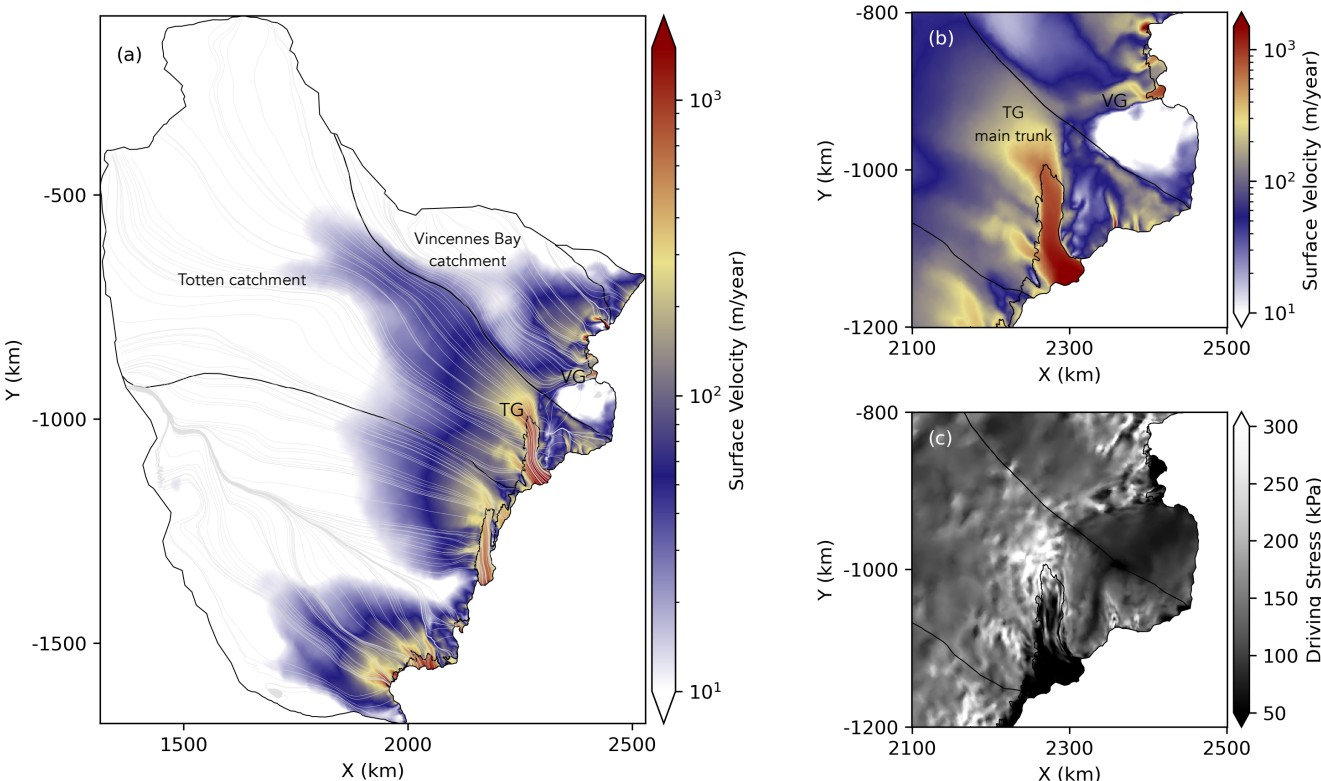

**Figure 2.** (a) and (b) Ice surface speeds (m year$^{-1}$) from MEaSUREs2; (c) driving stress $p$ (kPa). Black contours are as for Fig. 1. Light gray contours in panel (a) are velocity streamlines. Locations of the Totten Glacier (TG) and Vanderford Glacier (VG) are indicated.

by sedimentary basins, interpreted to represent deposition during rifting events that occurred well before the EAIS developed (Aitken et al., 2023). Sequences of these sedimentary rocks are stacked vertically, ranging from an older Neoproterozoic to lower Paleozoic sequence (Maritati et al., 2019), and a younger sedimentary sequence dating from the Permian-Triassic to the Cenozoic (Aitken et al., 2023). The topographic barriers of Highlands A, B and C, and the Knox Highlands typically comprise

erosional remnants of the older sequence, while the topographic basins typically comprise rocks of the younger sequence (Fig. 3).

The sedimentary cover in the main topographic lows reaches thicknesses of several kilometres, while the cover on the highlands is thinner. In regions of enhanced glacial erosion, the geology has been eroded to basement or possesses a mixture of basement with remnants of the younger sedimentary sequence (Aitken et al., 2016). Areas with sedimentary basins are in

general floored by more easily erodible rocks than in basement dominated regions, and upstream basins – particularly upstream of the main Totten tributary – are likely to provide an abundance of sediment to the downstream glacier bed, facilitating till-continuity and basal sliding (see section 2.4; Bell et al., 1998; Li et al., 2022).



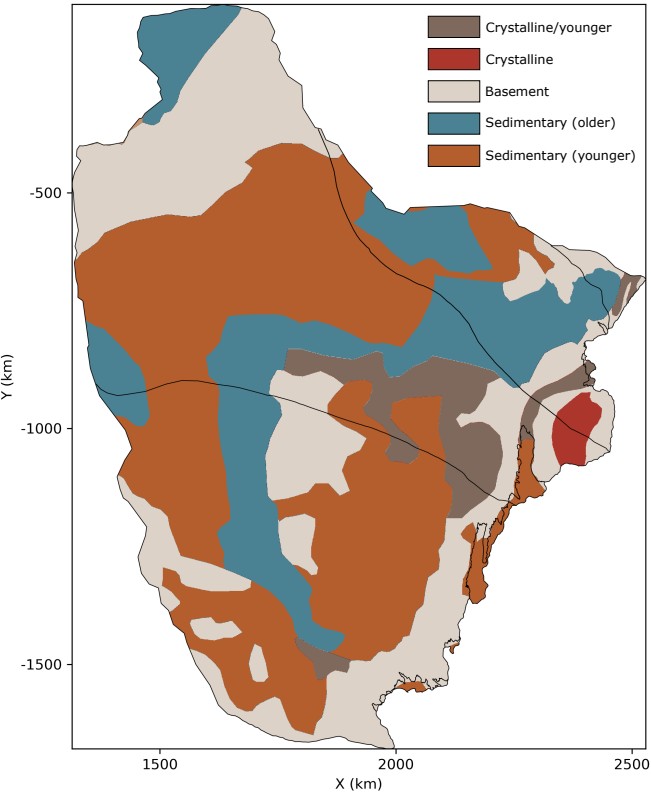

**Figure 3.** Interpreted bedrock geology of the ASB region showing the distribution of sedimentary rocks (Aitken et al., 2023). Variations in basement geology are not shown here.

Sedimentary basins also permit the occurrence of a viable hydrogeology system, where the permeable bed allows fluid exchange between the subglacial hydrological system and groundwater storage (Siegert et al., 2018). Groundwater discharge
into the glacier bed varies with ice sheet retreat rate and also with the thickness and permeability of the sedimentary basins, with major potential impacts on the basal sliding of contemporary glaciers in East and West Antarctica (Li et al., 2022). In the case of the Vincennes Bay catchment, the bed is dominated by the older sedimentary sequence and crystalline basement (Fig. 3), which are more likely to have low permeability; here, capacity for active hydrogeology may be restricted (although these regions of hard bed are characterised by channelised subglacial hydrology and faster flow; see sections 2.1 and 2.4). In
contrast, the ASB main basin – which is both thick and contains relatively young sedimentary rocks – has higher potential to support active hydrogeology, with potential impacts on basal sliding processes and hence ice streaming (Siegert et al., 2018; Li et al., 2022). Although estimates for the contributions of groundwater to the subglacial hydrology system exist broadly across Antarctica (Li et al., 2022), they are currently not available for the ASB specifically.





### 2.3 Bed topography

Approximately 75 % of the bed topography in the Totten and Vincennes Bay catchments is below sea level, and over 10 % is more than 1 km below sea level, particularly in the ASB main basin (Fig. 1). Regions of topographic highs play a role in determining the overall ASB catchment boundary as well as the orientation of flow within the Totten and Vincennes Bay catchments. This is notably the case with the topographic barriers of Highlands A and B, the western edge of the Vanderford Glacier, Law Dome, and the coastal ridge adjacent to Totten Glacier and the Moscow University Ice Shelf (Figs. 1, 3). Ice

flowing into the main Totten tributary has a similar flow trajectory to ice that flows into the eastern trunk of the Vanderford; ice from both glaciers originates in a region of generally high topography north of Lake Vostok (Vostok Highland; Fig. 1a). Downstream, both the Totten and Vanderford Glaciers flow into the Vanderford Trench – a deeply incised subglacial channel that borders Law Dome and records the lowest known topography of the ASB (-2,782 m below sea level; Fig. 1b). While topography provides a clear control on the overall ASB catchment boundary, there are no distinguishing features of the bed

topography that clearly control the location of the drainage divide between the Totten and Vincennes Bay catchments. However, possible local controls in the Vanderford Trench are difficult to observe where very steep topography and thick ice hinder imaging of the ice base by ice penetrating radar.

In terms of the susceptibility of each glacier to ongoing retreat, bathymetric pinning points on the Totten ice shelf, in the ~20 km downstream of the main Totten tributary grounding line, provide significant buttressing of the upstream glacier

(McCormack et al., 2021; Roberts et al., 2018). A prograde bed slope in the ~20 km upstream of the present-day grounding line decreases the likelihood of substantial retreat occurring under climate change projections to 2100 (Pelle et al., 2020; Sun et al., 2016). For the Vanderford Glacier, there is a high degree of uncertainty in the underlying bed topography within the Vanderford Trench; notably, uncertainties of 600 m and greater exist within the 10 km upstream of the present-day Vanderford grounding line (see discussion in Sect. 5 and Fig. 1c; Morlighem et al., 2020). Given the controlling influence of the bed

elevation and slope on grounding line retreat, this has significant implications for the observed and simulated timing and rate of retreat. These lines of evidence, coupled with the current rates of retreat from each glacier suggest that the Vanderford Glacier could retreat more rapidly, and further into the Vanderford Trench, than the Totten Glacier over the coming decades.

### 2.4 Subglacial hydrology and geomorphology

Observations and modelling confirm the presence of a widespread, persistent subglacial hydrological network in the ASB (Dow

et al., 2020), consisting of a distributed 'sheet-like' system in the ASB main basin and a channelised system in the few 100 km upstream of the Totten and Vanderford Glacier grounding lines (Fig. 4a). The proximal ice/bed interface of the distributed system is well lubricated and supports little to no basal shear stress (<20 kPa; Fig. 4b). Basal specularity content derived from radar is reduced within the channelised system (Dow et al., 2020), consistent with a rougher ice-bed interface where channels are present (Schroeder et al., 2013; Young et al., 2016), and a strong bed here supports most of the basal shear stresses for each

glacier (Fig. 4b). The calculation of the basal shear stresses in Fig. 4b is described in Sect. 4.1, and the striped pattern in the approach to the grounding line is consistent with previous studies (e.g. Sergienko and Hindmarsh, 2013).



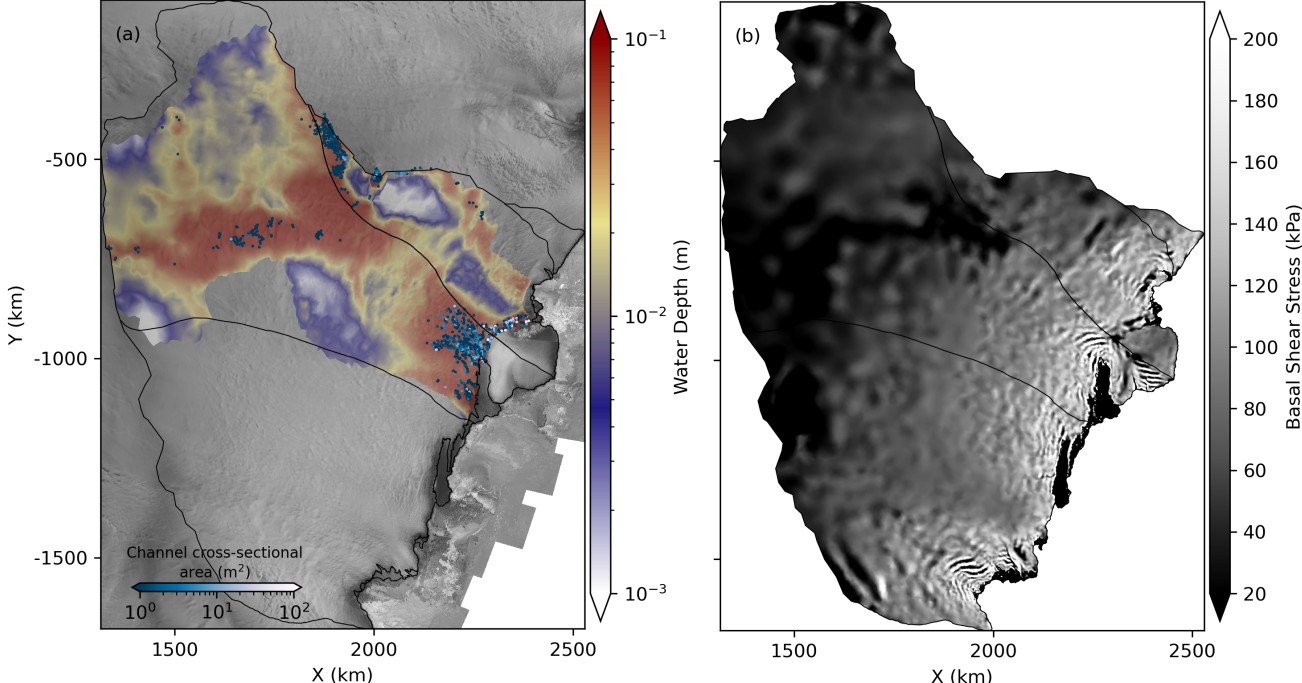

**Figure 4.** (a) Water depth (m) in the distributed and channelised subglacial hydrology system generated using GlaDS (Dow et al., 2020); and (b) basal shear stress (kPa) from the ice sheet model simulation, as described in Eq. (1) in Sect. 4.1. Black contours are the same as in Fig. 1.

The majority of basal water in the ASB is routed towards the Totten Glacier (Dow et al., 2020; Wright et al., 2012; Young et al., 2016): output from the Glacier Drainage System (GlaDS) subglacial hydrology model estimates approximately $42\,\mathrm{m^3\,s^{-1}}$ discharge from the Totten catchment into the ocean, compared with only $3\,\mathrm{m^3\,s^{-1}}$ from the Vanderford (Dow et al., 2020). This

is partly due to fewer channels developing in the approach to the Vanderford grounding line in the steady-state GlaDS simulations compared to the Totten grounding line. Nevertheless, the subglacial hydrological networks of the Totten and Vincennes Bay catchments are strongly connected. For example, ICESat surveys identified at least two 'active' lakes located within the $200\,\mathrm{km}$ upstream of the Totten grounding line, with lake drainage and filling evidenced in surface elevation changes between 2003 and 2006 (Smith et al., 2009). Hydraulic potential analysis of one of these lakes shows that basal water drainage may be

routed from the Totten to the Vanderford Glacier with only small changes in the ice surface elevation (Young et al., 2016).

     The geomorphology of the continental shelf proximal to the Vanderford Glacier – revealed in high resolution multibeam bathymetry (Commonwealth of Australia, 2022) – provides evidence for an active subglacial hydrology network in the Vincennes Bay region in the past. Of particular note is the presence of deeply incised bedrock channels, which are the most abundant landform on the inner continental shelf, and which span depths of $450\,\mathrm{m}$ to $2280\,\mathrm{m}$ below sea level. Similar bedrock

channels and flat-bottomed basins are also observed in front of the Thwaites and Pine Island Glaciers in West Antarctica (Lowe





and Anderson, 2003; Nitsche et al., 2013), where hydrological modelling suggests that these are relict subglacial meltwater channels and lakes (Beaud et al., 2018; Kirkham et al., 2019). The presence of these landforms on the inner shelf of Vincennes Bay supports the existence of an active, dominant subglacial hydrological system beneath a more extensive Vanderford Glacier, and which formed over multiple glacial periods. A previous study (Kirkham et al., 2019) has also documented a series of relict

subglacial lakes akin to those mapped in front of Vanderford (indicated in gray in Fig. 5). The network of subglacial meltwater channels and lakes would have formed during the last glacial period, or possibly over multiple glacial periods (Kirkham et al., 2019; Beaud et al., 2018). Based on lake geometry and contemporary subglacial water transfer, these lakes likely drained episodically through the subglacial meltwater channels on decadal timescales (Kirkham et al., 2019), facilitating ice basal sliding over the rugged topography of the inner continental shelf.

In summary, the large-scale geometry of the ASB controls the current divergence of flow between the Totten and Vanderford Glaciers. However, it is likely that the characteristics of, and variability in, the subglacial environment, particularly those elements that influence basal motion (e.g. the bed substrate and deformability; subglacial hydrological network; groundwater system) in the ASB, play a key role in the flow configuration. As basal water pressure and water accumulation vary into the future – e.g. associated with drawdown of the Vanderford Glacier due to grounding line retreat and subsequent thinning – basal

water and ice flow piracy towards the Vanderford may also occur.

## 3 Climate drivers of the ASB

### 3.1 Atmospheric drivers

Surface mass balance (SMB) is the primary atmospheric driver of variability between the Totten and Vanderford Glaciers. The atmospheric circulation over Law Dome is predominately polar easterlies (Bromwich, 1988) and the orography of the

Dome causes relatively large snow accumulation, with a strong east-west gradient (Morgan et al., 1997; Udy et al., 2021). The temporal distribution of the snow accumulation is related to the passage of synoptic scale features, and is therefore episodic, typical of coastal Antarctica (Bromwich, 1988).

Antarctic snow accumulation is expected to increase as water vapour increases with a warming atmosphere (Krinner et al., 2007). While there is no clear trend in Law Dome accumulation rates over the past 2000 years (Roberts et al., 2015), a trend of

increasing total Antarctic annual snow accumulation has been observed since 1800 (Thomas et al., 2017). While high-spatial resolution snow accumulation observations are broadly lacking, the RACMO2.3p2 reanalysis model (van Wessem et al., 2018) captures the strong east-west gradient in snow accumulation over Law Dome on annual time-scales. Results averaged over Totten and Vanderford Glaciers (Fig. 6) suggest that without major changes in atmospheric circulation patterns, a local increase in snow accumulation over Law Dome will amplify the east-west gradient.

In addition to the direct snow accumulation implications, there is a secondary effect on glacier flow related to the enhanced vertical advection of cool surface temperature into the interior at increased snow accumulation rates. For pseudo steady-state surface elevations and approximately equal surface temperatures, the higher snow accumulation over the Totten will result in higher vertical velocities compared to the Vanderford. Without compensating differences in geothermal heat flow,



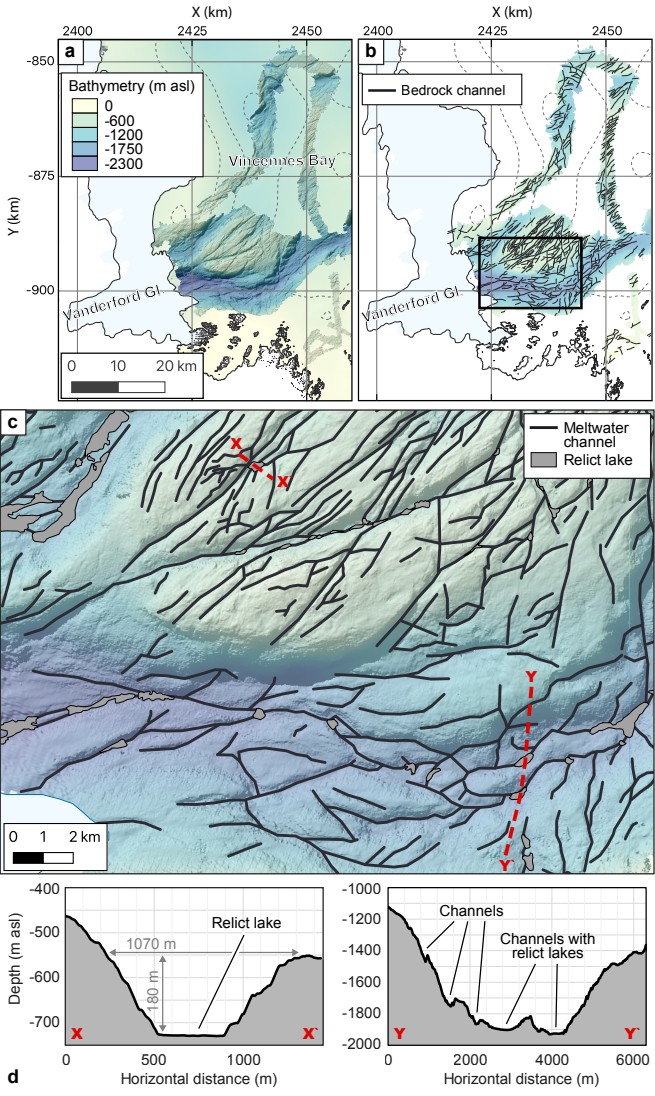

**Figure 5.** Geomorphology of the continental shelf in front of Vanderford Glacier. (a) High-resolution multibeam bathymetry (Commonwealth of Australia, 2022) highlighting a submarine canyon, overlaid on previous bathymetric estimates (Arndt et al., 2013; Morlighem et al., 2020) with contours at 250 m intervals. (b) Bedrock channels (black lines); black square shows the location of (c). (c) Bedrock channels of the inner continental shelf, interpreted as relict subglacial meltwater channels; flat-bottom parts of these channels (slope $\leq 2°$) are interpreted as relict lakes. (d) Geometry of the meltwater landforms highlighted in cross-section profiles. Channels follow lines of geological weakness in shallow and deep areas (from 450 m to 2280 m water depth). A large bedrock high has major channels (150 to 280 m deep, 1 km wide and 9 to 16 km long) connected by shorter, less-incised channels. Within the two troughs, bedrock channels have an anastomosing pattern, with meandering central channels linked in multiple places to neighbouring channels. Flat surfaces are found in deeper parts of the main channels, reflecting localised sedimentation.




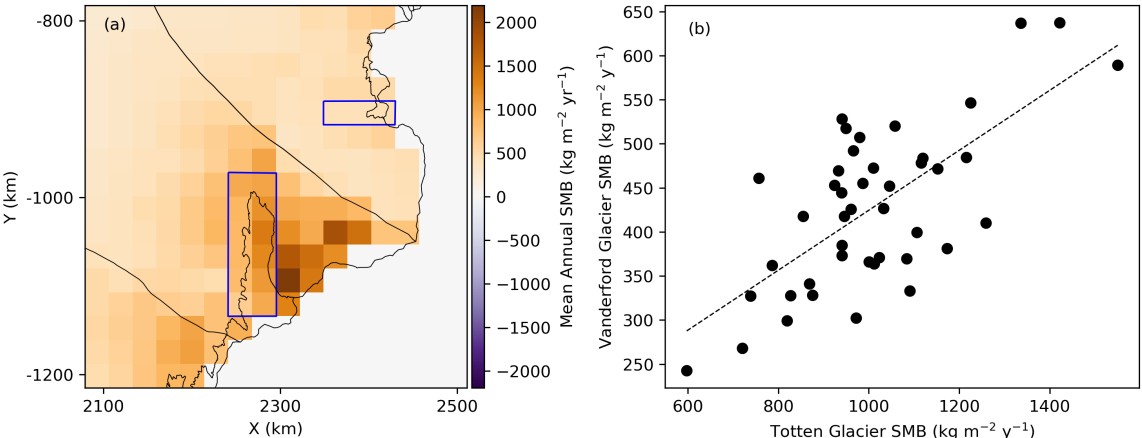

**Figure 6.** Surface mass balance over Law Dome and surrounding area. a) RACMO2.3p2 reanalysis (van Wessem et al., 2018) averaged over the period 1979 to 2021, showing a strong east-west snow accumulation gradient over Law Dome. Totten (larger) and Vanderford (smaller) bounding boxes for the SMB analysis in panel b are shown (blue). b) Vanderford annual snow accumulation as a function of Totten annual snowfall. Least-squares linear regression line shows slopes less than unity, indicating that for increasing local snow accumulation the mismatch between the Totten and Vanderford SMB is expected to increase.

deformational heating or heat transport from basal hydrology processes, this increased vertical advection of cold will result in

the bulk ice properties of the Totten being colder and stiffer than the Vanderford.

Without considering the effects of ocean-driven ice shelf melting (and associated surface lowering), the projected SMB changes in the ASB under a warming climate could lead to enhanced surface elevation and thickness increases at the Totten Glacier compared with the Vanderford Glacier. These effects could be further enhanced if they were to occur in concert with a dynamics-driven deceleration of the Totten Glacier due to cooling and stiffening ice over timescales of centuries to millennia.

**3.2  Oceanic drivers**

Similarly to glaciers in the Amundsen Sea Sector of West Antarctica, ice shelf thinning, grounding line retreat and increased discharge at Totten Glacier is linked to the incursion of warm modified circumpolar deep water (mCDW) through deeply incised bathymetric channels (Greene et al., 2017; Roberts et al., 2018; Silvano et al., 2018, 2019), evidenced from both airborne and ship-borne measurements (Greenbaum et al., 2015; Rintoul et al., 2016). Ice shelf melt rates at the Totten Glacier have been

estimated to exceed $50\,\mathrm{m\,year^{-1}}$ near the grounding line (Adusumilli et al., 2020), and are among the largest observed melt rates in East Antarctica. Modelling indicates increased melt in recent decades, attributed primarily to freshening shelf water (Nakayama et al., 2021). This is despite a consistent trend of polewards migration and warming of mCDW off-shelf (Herraiz-Borreguero and Naveira Garabato, 2022; Yamazaki et al., 2021), which is projected to increase as the climate warms. However,





observational evidence and modelling outputs are consistent with significant variability in melt rates as a result of variable
mCDW supply in this region (Gwyther et al., 2014, 2018; Nakayama et al., 2021; Paolo et al., 2015; Silvano et al., 2019).

Links between ocean forcing and recent trends at Vanderford Glacier are less well established than at Totten Glacier. Data
from seal dives suggest that the mCDW in Vincennes Bay is up to 0.5°C and the warmest sampled to date in East Antarctica;
however, temperatures at Vanderford Glacier ice shelf of -0.5°C are comparable to those at the Totten Glacier ice front of
-0.4°C (Ribeiro et al., 2021). The presence of a deep bathymetric trough at the Vanderford ice front (Commonwealth of
Australia, 2022) provides a clear pathway for mCDW incursion to the ice shelf cavity, although the bathymetric connection
from this trough to the continental shelf slope has not been mapped. It is hypothesised that the ocean is driving the observed
retreat and thinning of the Vanderford Glacier 2003 to 2020 (Picton et al., 2022), but observations and modelling are needed to
establish causal relationships.

Modelling has been used to evaluate the potential for enhanced melting in the Sabrina and Knox Coasts over the coming
centuries (Sun et al., 2016). Upwelling of mCDW is positively correlated with the Southern Annular Mode (SAM), and positive
SAM trends are projected to continue through the 21st Century, which could lead to enhanced mCDW-driven ice shelf melt
(Herraiz-Borreguero and Naveira Garabato, 2022; Purich and England, 2021). Surface winds near the East Antarctic coastal
margin are projected to intensify under climate warming (Fyfe et al., 2007; Spence et al., 2014; Wang, 2013), which could
also increase wind-driven upwelling of mCDW to the continental shelf, and the supply of warm water to ice shelf cavities
(Greene et al., 2017). Naughten et al. (2018) suggested that warming surface waters linked to a decline in summer sea ice cover
could dominate freshening associated with ice sheet mass loss trends in this sector. A key caveat on modelling results is the
generally low spatial resolution (hence coarsely-resolved bathymetry) used and the fact that previous bathymetry products do
not incorporate the deep bathymetric trough in Vincennes Bay (Commonwealth of Australia, 2022) important for warm water
supply to the ice shelf cavity. Nevertheless, evidence consistently points to the possibility of increasing ocean-driven ice shelf
melt in this sector.

The cumulation of these factors – the relatively warmer mCDW in Vincennes Bay, the pathway for its incursion into the
Vanderford ice shelf cavity, and the greater potential for grounding line retreat of Vanderford Glacier into the Vanderford
Trench compared to the buttressed Totten Glacier – suggest a potential heightened vulnerability of the Vanderford Glacier to
continued thinning in the coming decades to centuries.

**4   Sensitivity of driving stress and hydraulic potential to changes in surface elevation**

Here, we use the Ice-sheet and Sea-level System Model (ISSM; Larour et al., 2012) to quantify the impact on ice and basal
water routing due to ice surface elevation changes at Totten and Vanderford glaciers that are consistent with present-day flow
dynamics and address the potential climate scenarios discussed in Sect. 3. That is, we generate three ice surface elevation
perturbations, based on: (1) an increase in SMB in the Totten catchment (defined as expSMB); (2) a decrease in basal friction
in the Vincennes Bay catchment (defined as expFriction); and (3) a combination of (1) and (2) (defined as expCombined). The
model setup and experiments are discussed in more detail below.



## 4.1 Model setup and experiments

The model domain covers the ASB (Fig. 1a) and comprises 58,938 anisotropic mesh elements. The initial bed topography, thickness, surface elevation and ice masks are from BedMachine Antarctica v3 (Morlighem et al., 2020), and surface veloc-
ities from MEaSUREs version 2 (MEaSUREs2 Rignot et al., 2011, 2017). Using the Shallow Shelf Approximation (SSA; MacAyeal, 1989) and inverse methods, we calculate the basal friction coefficient across the grounded ASB using the Budd friction law (Budd et al., 1979), given by

$$\tau_b = C_f^2 N u_b, \tag{1}$$

where $\tau_b$ (Pa) is the basal shear stress, $C_f$ ($s^{1/2}\,m^{-1/2}$) is the friction coefficient, and $u_b$ (m year$^{-1}$) is the basal sliding velocity.
The effective pressure $N$ (Pa) is given by $N = \rho_{ice}gh + \rho_{water}gb$, where $\rho_{ice}$ is the ice density (kg m$^{-3}$), $g$ is the gravitational acceleration (m s$^{-2}$), $h$ is the ice thickness (m), $\rho_{water}$ is the density of freshwater (kg m$^{-3}$), and $b$ is the bed elevation (m). We assume a Glen-type flow relation (Glen, 1953), where the viscosity $\mu$ (Pa s) is given by

$$\mu = \frac{B}{2\dot{\varepsilon}_e^{1-n/n}}, \tag{2}$$

and inverse methods over the entire domain to calculate the ice rigidity $B$ (Pa s$^{1/n}$), for effective strain rate $\dot{\varepsilon}_e$ (s$^{-1}$) and $n = 3$.
The inverse method relies on minimising a cost function that includes terms for the linear and logarithmic misfit between the simulated and observed surface velocities, as well as regularisation parameters that smooth strong gradients in the rigidity and basal friction coefficient (Morlighem et al., 2013). MEaSUREs2 velocities are used at the inflow boundaries and a free flux boundary condition is applied at the calving front.

Holding the grounding line positions fixed, we run a 200 year spin-up simulation with monthly time stepping, at the end of
which the ice surface speed and geometry is in a pseudo steady-state. For this simulation we use annual average surface mass balance from RACMO2.3p2 and basal melt rates are calculated using a parameterisation that linearly decreases the basal melt from 30 m year$^{-1}$ at ice shelf drafts of 400 m below sea level or deeper, to 5 m year$^{-1}$ at ice shelf drafts of 200 m below sea level or shallower (Seroussi et al., 2017). The final ice velocities, surface elevation and thicknesses from this simulation are used as the initial conditions for our perturbation experiments described below.
To generate ice surface elevation increases at Totten Glacier (expSMB), we increase the annual average SMB as follows:

$$SMB_{new} = SMB_{racmo} + \beta SMB_t, \tag{3}$$

where $SMB_{new}$ is the perturbation field, $SMB_{racmo}$ is the observed annual average SMB from RACMO2.3p2 over the period 1979 to 2021 (van Wessem et al., 2018), and $SMB_t$ is the observed annual average SMB within the Totten catchment and zero SMB elsewhere in the ASB. Values for the constant $\beta$ increase from 0.06 to 0.42, in increments of 0.06, and are chosen to
represent the percentage increases in SMB that correspond to 1°C increases in air surface temperature from 1 to 7°C (Frieler et al., 2015). Although we note that other surface processes influence the SMB, we assume that increases in accumulation will dominate at basin-scale.



To generate ice surface elevation decreases at Vanderford Glacier (expFriction), we decrease the basal friction coefficient within the Vincennes Bay catchment as follows:

$$C_{new} = C_f - \sigma C_v M^{\alpha}, \tag{4}$$

where $C_{new}$ is the perturbation field, $C_f$ is the basal friction coefficient calculated using inversion, and $C_v$ is the basal friction coefficient within the Vincennes Bay catchment and zero elsewhere in the ASB. Values for the constant $\sigma$ increase from 0.1 to 0.7 in increments of 0.1. These values are chosen to yield the same area-weighted percentage changes between the simulated surface elevation compared with the control, when averaged over the Vincennes Bay catchment, as the corresponding surface elevation change generated from expSMB. $M$ is a mask that is generated by calculating the distance from the grounding line within the Vincennes Bay catchment, and then normalising the field to vary linearly from 1 at the grounding line to 0 at the furthest point (on the southern inflow catchment boundary). We choose $\alpha = 20$ as the exponent to $M$, which concentrates the change in the basal friction coefficient within the $\sim$100 km upstream of the Vanderford Glacier grounding line (see appendix A). We reduce the basal friction coefficient in the Vincennes Bay catchment rather than modify the basal melt rate because, although not directly observable, we expect a reduction in basal traction as a consequence of increased ocean melting, retreat, and acceleration. In this way, we avoid increasing the basal melt rates at Vanderford Glacier relative to Totten Glacier (given the lack of evidence for such a relative increase in ocean forcing), while still capturing the effects of continued retreat at Vanderford Glacier, which is more likely than at Totten Glacier over the coming decades to century (Sect. 3). We also emphasise that the aim here is not to model the expected evolution of this system, but to generate surface change fields that are dynamically consistent with increasing SMB at Totten Glacier and continued retreat at Vanderford Glacier, to ascertain the impact of perturbations in the surface elevation on ice and basal water flow piracy.

In addition to the expSMB and expFriction experiments, a third experiment (expCombined) combines the SMB and basal friction coefficient fields in Eqs. 3 and 4. Using the output geometries (thickness and surface elevation) and velocities from the pseudo steady-state simulation, we run each of the perturbation experiments expSMB, expFriction and expCombined for 1,000 years with monthly time stepping. A 1,000 year control run that uses the original $SMB_{racmo}$ and $C_f$ fields is also simulated. Figure 7 shows the differences between the perturbed surface elevations and the control ($dS_i$, where the subscript $i$ refers to $SMB$, $F$ or $C$) for an air surface temperature increase of 1°C (i.e. for $\beta = 0.06$ in Eq. (3) and $\sigma = 0.1$ in Eq. (4)); hereafter denoted expSMB$_1$, expFriction$_1$ and expCombined$_1$.

## 4.2 Driving stresses and ice flow routing

We investigate the impact of the changed surface elevations on the driving stresses and ice flow routing within the ASB. The driving stress $p$ (Pa) is given by:

$$p = \rho_{ice} g h \nabla S_i, \tag{5}$$

where $\nabla S_i$ is the gradient in the perturbed surface elevation – the sum of the BedMachine surface elevation and each $dS_i$ field.

For expSMB$_1$, the driving stress decreases through much of the interior of the ASB, but increases within $\sim$10 km of the Totten Glacier grounding line. This is consistent with decreasing velocities upstream of the Totten Glacier grounding line, but



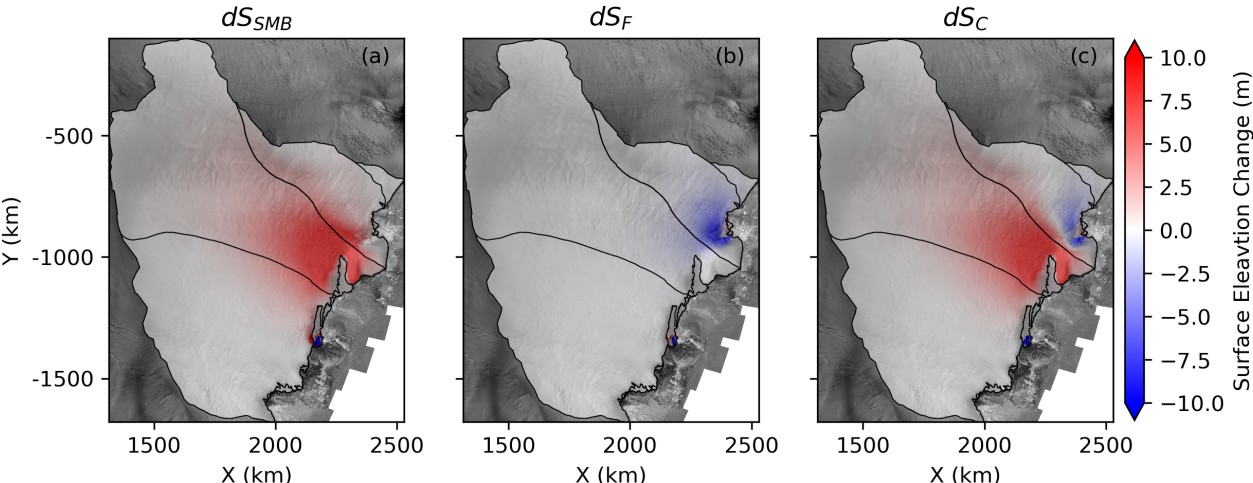

**Figure 7.** Surface elevation changes ($dS_i$) in the perturbation experiments compared with the control (m) for the $1^\circ$C increase. (a) $dS_{SMB}$ (expSMB$_1$), (b) $dS_F$ (expFriction$_1$), and (c) $dS_C$ (expCombined$_1$).

**Table 1.** Absolute and percentage changes in the Vincennes Bay catchment areas at the end of the 1000 year perturbation experiments. The catchment area defined by the MEaSUREs2 velocities is $71{,}329\,\text{km}^2$ (compared with $68{,}902\,\text{km}^2$ defined by IMBIE).

|  | expSMB | | expFriction | | expCombined | |
|---|---|---|---|---|---|---|
|  | abs (m) | % | abs (m) | % | abs (m) | % |
| $1^\circ$C | 1,208 | 1.7 | 958 | 1.3 | 2,122 | 3.0 |
| $2^\circ$C | 2,737 | 3.8 | 1,226 | 1.7 | 2,961 | 4.2 |
| $3^\circ$C | 4,692 | 6.6 | 1,181 | 1.7 | 6,493 | 9.1 |
| $4^\circ$C | 6,695 | 9.4 | 1,152 | 1.6 | 8,633 | 12 |
| $5^\circ$C | 8,073 | 11 | 1,865 | 2.6 | 12,324 | 17 |
| $6^\circ$C | 11,657 | 16 | 2,259 | 3.2 | 14,779 | 21 |
| $7^\circ$C | 12,104 | 17 | 2,885 | 4.0 | 15,890 | 22 |

increasing velocities on the Totten Glacier ice shelf, and Vanderford Glacier and ice shelf. The divide between the Totten and Vincennes Bay catchments migrates eastwards for expSMB$_1$ compared with the catchments defined using the MEaSUREs2 velocities, increasing the Vincennes Bay catchment area by $\sim$1,208 km$^2$ (1.7 %; table 1). This leads to a rerouting of ice flow from the Totten Glacier towards the Vanderford Glacier, with an increase in discharge from the Vanderford Glacier of $\sim$0.02 Gt year$^{-1}$ (i.e. 1.6 % increase; table 2). Both trends increase with increasing air surface temperature change; for expSMB$_7$, the Vincennes Bay catchment area increases by 17 % and the ice discharge increases by 11 %.



**Table 2.** Absolute and percentage changes in the Vanderford Glacier discharge (Gt year$^{-1}$) at the end of the 1,000 year perturbation experiments. Differences are with respect to the values calculated using MEaSUREs2 velocities.

|  | expSMB | | expFriction | | expCombined | |
|---|---|---|---|---|---|---|
|  | abs (Gt year$^{-1}$) | % | abs (Gt year$^{-1}$) | % | abs (Gt year$^{-1}$) | % |
| 1°C | 0.02 | 1.6 | 0.05 | 5.1 | 0.07 | 6.8 |
| 2°C | 0.03 | 3.2 | 0.10 | 11 | 0.14 | 14 |
| 3°C | 0.05 | 4.8 | 0.16 | 16 | 0.21 | 22 |
| 4°C | 0.06 | 6.3 | 0.22 | 22 | 0.29 | 30 |
| 5°C | 0.08 | 7.9 | 0.28 | 28 | 0.37 | 38 |
| 6°C | 0.09 | 9.5 | 0.34 | 35 | 0.46 | 47 |
| 7°C | 0.11 | 11 | 0.41 | 42 | 0.56 | 57 |

Decreasing the friction coefficient in the Vincennes Bay catchment (expFriction$_1$) leads to immediate acceleration of the Vanderford Glacier and surface elevation lowering upstream of the grounding line (Fig. 7) The driving stress also decreases close to the grounding line and on the Vanderford Glacier ice shelf, although increases upstream of the grounding line to
the Elcheikh saddle point (Fig. 8). This also results in an eastwards migration of the divide (i.e. into the Totten catchment) between the Totten and Vincennes Bay catchments, increasing the Vincennes Bay catchment area by 958 km$^2$ (1.3 %) – less of a migration than observed in expSMB$_1$. By contrast, the flux across the Vanderford Glacier grounding line increases by 5.1 %, which is over three times that of expSMB$_1$, reflecting the relatively greater increase in velocities in the Vincennes Bay catchment. For expFriction$_7$, the discharge increases by 42 %, reflecting the greater sensitivity to relative increases in ice
surface speed.

The driving stress changes for expCombined$_1$, along with the changes in the Vincennes catchment areas and discharges, are close to a linear combination of expSMB$_1$ and expFriction$_1$ (Fig. 8). The magnitude of the differences increases for increasing air surface temperature, such that in expCombined$_7$, the increase in the Vincennes Bay catchment area is ∼5 % larger than expSMB$_7$, and the discharge 15 % larger than in expFriction$_7$.

**4.3 Basal water routing**

We next examine the sensitivity of basal water routing between the Totten and Vanderford Glaciers due to changes in the surface elevation. We use the Matlab TopoToolbox (Schwanghart and Scherler, 2014) to calculate the basal water accumulation across the ASB (the total upstream area in km$^2$ that contributes to water accumulation within each individual grid cell). The upstream area is defined based on the basal water flow direction vector, which is calculated using the local slope in the effective pressure
at the base of the ice sheet (Greene et al., 2017), and an upstream cell can provide water to only one neighbouring outlet cell. Here, rather than assuming overburden hydraulic potential, we use the parameterisation for the effective pressure $N_E$ given by



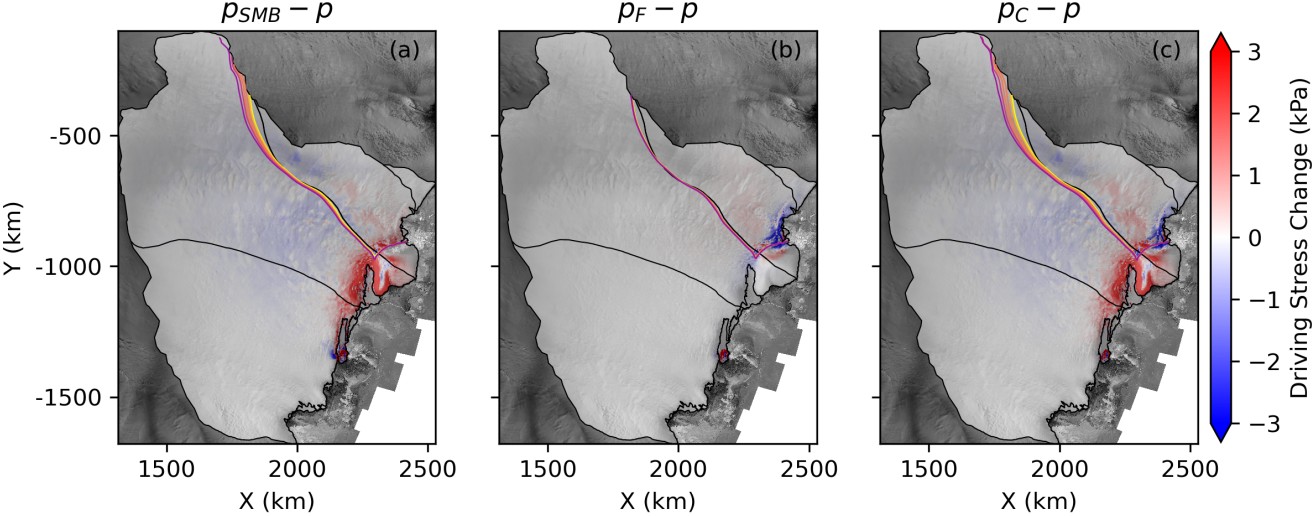

**Figure 8.** Driving stress changes (kPa) in the perturbation experiments for a $1°C$ increase compared with the control (m). (a) $dS_{SMB}$ (expSMB$_1$), (b) $dS_F$ (expFriction$_1$), and (c) $dS_C$ (expCombined$_1$). The contours are the changes in the catchment divides for each of the perturbation experiments for 1 to $7°C$, coloured brightest to dullest (yellow to purple), respectively.

McArthur et al. (2023), defined as:

$$N_E = \rho_{ice}gh_i(1-r_l)\frac{\tilde{h_i}^m}{\tilde{h_i}^m + h_i^m}, \tag{6}$$

where $r_l$, $\tilde{h}$ and $m$ are constants, and the fields $h_i$ are calculated by summing the BedMachine thickness and the perturbed

thickness from each simulation. A summary of the derivation of this parameterisation is provided in the appendix B.

The basal water accumulation calculated using the BedMachine bed topography, thickness and surface elevation is shown in Fig. 9a. Currently, basal water is preferentially routed towards Totten over any other glacier in the ASB. This pattern of basal water accumulation is very similar for each of the surface change experiments under 1 to $4°C$ warming. However, for the expSMB$_5$ and expCombined$_5$ experiments, basal water piracy to Vincennes Bay occurs, with the glaciers in this sector draining

almost the entire region in the interior of the ASB (Fig. 9b,d). Under these scenarios, the only water routing towards the Totten Glacier originates north of Highland B. The pattern is consistent, with little subsequent change, for air surface temperature increases above $5°C$. By contrast, basal water accumulation is relatively insensitive to changes in surface elevation associated with the basal traction reduction over the Vanderford catchment imposed in our experiments, over the entire investigated range (i.e. expFriction$_1$ to expFriction$_7$; Fig. 9c).





**Figure 9.** Basal water accumulation (km$^2$) using (a) BedMachine geometries ($A_{obs}$); (b) geometry fields from expSMB$_5$ ($A_{SMB}$); (c) geometry fields from expFriction$_5$ ($A_F$); and (d) geometry fields from expCombined$_5$ ($A_C$). The black square represents the switch point of changes in water routing between the different experiments.



## 5 Discussion and summary

This study examined the potential for ice flow and basal water piracy between the Totten and Vincennes Bay catchments due to changes in the ice surface elevation induced by thinning at Vanderford Glacier (e.g. due to ongoing grounding line retreat) and increased SMB at Totten Glacier. For each scenario of ice surface elevation change, irrespective of the corresponding degree of warming, a rerouting of ice flow from the Totten Glacier towards the Vanderford Glacier occurred, with subsequent increases in the ice discharge at Vanderford. Increasing the SMB led to a greater eastwards migration of the boundary between the two catchments than decreasing the basal traction, although the discharge towards Vanderford increased more substantially with increases in the basal friction coefficient. There was no amplification in the response when the two effects were combined, with increases in the discharge and Vincennes Bay catchment area towards the Vanderford that were essentially a linear superposition of the two separate effects.

Very little change was observed in the basal water routing for surface elevation changes corresponding to air surface temperature increases up to 4°C. However, surface elevation increases in the expSMB$_5$ and expCombined$_5$ scenarios led to a re-routing of basal water from Totten Glacier to the Vincennes Bay glaciers. This effect persisted under increasing air surface temperatures, with no subsequent change in routing. Interestingly, this effect was not observed in the cases of thinning at Vanderford Glacier, even in the most extreme case (expFriction$_7$). The switch in basal water routing from the Totten to the Vanderford Glacier was sudden, with otherwise relatively small variations in the overall basal water routing calculations. In all scenarios, the switch occurred within a region where the basal water routing from the ASB main basin converged, before diverging again to the Totten and Vincennes Bay catchments (as indicated in the black box, Fig. 9). The median ice surface increase from the expCombined$_4$ to expCombined$_5$ experiments compared with the control was 3.1 m in this region (i.e. from 12.4 m in expCombined$_4$ to 15.5 m in expCombined$_5$). Although this particular "switch" location may differ in our calculations using the hydraulic potential from those using a subglacial hydrology model, this result may indicate the importance of considering regions of flow convergence in the identification of tipping points in basal water routing.

The surface elevation changes generated in expFriction, and consequent impacts on ice and basal water routing, were strongly impacted by both how far inland the effect of reduced friction penetrated and the magnitude of the perturbation itself. Due to the limited region over which the perturbation in basal friction was concentrated, changes in the surface elevation far upstream were relatively minor compared with those resulting from thickening simulated at Totten Glacier, limiting the corresponding magnitude of changes in the ice and basal water piracy. Hence, a greater response might occur if the thinning were to penetrate deeper into the ASB.

Our analysis relied on parameterised hydraulic potential to infer changes in basal water routing, which is not as robust as using a subglacial hydrology model that incorporates the full physics (Dow, 2019). For example, the growth of channels will impact the local hydraulic potential gradients – an effect which is neglected in this study. Dow (2019) also showed that the basal water routing between the Totten and Vanderford glaciers is highly sensitive to details in the bed topography elevations, such that uncertainties in the bed topography may influence calculations based on hydraulic potential. Furthermore, ice sheet-hydrology interactions could influence the sensitivity of ice and basal water routing, even under more minor changes





in the overlying ice sheet geometry than those found here. If basal water piracy were to occur as Vanderford Glacier retreats,
enhanced basal lubrication at Vanderford could cause acceleration beyond what is currently predicted by many stand-alone
ice sheet models (Seroussi et al., 2020). On the other hand, steeper surface elevation slopes in the Vanderford Glacier could
result in larger and more efficient subglacial hydrology channel formation, which could cause a reduction in ice speed near the
grounding line due to lower basal water pressures (Alley et al., 1994; Dow et al., 2022). Our analysis excluded the influence of
variable melt rates (due to frictional or geothermal heat flow, e.g. McCormack et al., 2022) and changes in the groundwater flux
due to loading and unloading (Li et al., 2022; Siegert et al., 2018), both of which impact the supply of water, as well as the flow
routing. We also neglected basal hydro-thermomechanical processes, which impact overall ice flow dynamics (Arthern et al.,
2015; Sergienko and Hindmarsh, 2013), and which have been shown to play a role in redistributing meltwater and impacting
the strength of the bed in the Siple Coast ice streams (Bougamont et al., 2015). Uncertainty associated with these competing
processes and the exact thresholds under which we might expect basal water piracy to occur motivates further investigation
into coupled hydro-thermomechanical interactions and dynamics in the ASB.

Our numerical experiments were designed to simulate only the effects of increased surface elevation at Totten Glacier and
decreased surface elevation at Vanderford Glacier, and did not include the competing influence of thinning at Totten Glacier
nor ongoing grounding line retreat at Vanderford Glacier. Satellite observations to 2012 have shown a lowering of the Totten
Glacier surface elevation (Picton et al., 2022; Smith et al., 2020), with estimates of $1.2\pm0.6$ m year$^{-1}$ (Flament and Remy,
2012), linked to ocean-driven thinning of the ice shelf and consequent dynamic thinning of the upstream glacier (Li et al.,
2016; Rignot et al., 2019; Roberts et al., 2018). Various ice sheet models predict continued thinning at Totten Glacier under
climate change scenarios into the future (Seroussi et al., 2020), which could outweigh gains in surface elevation from increased
SMB. Nevertheless, the main Totten tributary is topographically constrained (Pelle et al., 2020), currently limiting its retreat
beyond the Vanderford Trench ridge and into the upstream ASB main basin. By contrast, the Vanderford Glacier is currently
the fastest retreating glacier in EAIS and is retreating into a deepening subglacial bed (Morlighem et al., 2020). We did not
include the effect of grounding line retreat at Vanderford Glacier on the surface elevation changes generated here, although
grounding line retreat is likely to compound the thinning simulated by our model. It is unclear how the rates of thinning at
Totten and Vanderford glaciers will evolve into the future and it is possible that proportionally higher thinning at Vanderford
than Totten could be sufficient to divert flow to Vanderford Glacier.

There are a number of key uncertainties in how rapidly these glaciers will retreat into the future – including the impact of
retreat into the marine basin of the ASB, as has been evidenced during past warm periods (Aitken et al., 2016; DeConto and
Pollard, 2016; Golledge et al., 2015). First, detailed knowledge of the subglacial topography is essential in accurately predicting
retreat rates, both in the ASB (McCormack et al., 2021) and elsewhere in Antarctica (Castleman et al., 2022; Schlegel et al.,
2018; Seroussi et al., 2020), and the sensitivity of basal water routing to relatively small changes in the topography (Dow, 2019).
There are relatively high uncertainties in ASB topography, and particularly where there is: (1) a paucity of radar transects, e.g.
in the ASB main basin and basal water "switch" region (black box; Fig. 9); (2) relatively slow-flowing ice, e.g. in the interior of
the ASB where mass conservation estimates of bed topography are less reliable; and (3) deep topography, where side reflections
are strong and mask bed returns (Fig. 1d). For the latter, this is particularly the case in Vanderford Glacier, upstream of the



grounding line. Improved knowledge of the bed geometry and sediment characteristics in these regions is essential, and should be a focus of future airborne surveys.

Second, SMB impacts the long-term evolution of the ice sheet geometry, yet projections in this region are highly uncertain. This is largely because Coupled General Circulation Models used to project changes in accumulation are relatively coarse resolution (typically ∼100 km) – too coarse to accurately capture topographic or other effects that are essential in modelling Antarctic accumulation (e.g. Ghilain et al., 2022). This is likely to influence SMB projections, particularly in the region of high accumulation at Totten Glacier, and hence predictions of whether, or to what degree, surface increases due to SMB could compensate for surface lowering due to ocean-driven thinning.

Finally, while links have been made between the presence of mCDW in the Vincennes Bay region (Ribeiro et al., 2021) and rapid retreat of the Vanderford Glacier in the past few decades (Picton et al., 2022), a causal relationship between ocean drivers and thinning and retreat has not been established. This is partly due to the relatively coarse spatial resolution of melt rates beneath the Vanderford Glacier ice shelf generated from satellite observations, which are also sparsely sampled in time (Greene et al., 2022), and a lack of consistent ocean state measurements in Vincennes Bay. Given that accurate knowledge of ocean melt rates is critical for ice sheet model simulations of retreat, and the evidence for a strong observed warming trend of mCDW in this region of East Antarctica (Herraiz-Borreguero and Naveira Garabato, 2022), this emphasises the need for long-term ocean state monitoring in this region.

## 6 Conclusions

Coupled ice sheet-hydrology interactions may be at the heart of substantial ice and basal water reconfigurations in the past. For the Siple Coast ice streams, such reconfigurations have had a lasting impact on ice sheet mass balance: although the Kamb ice stream shutdown occurred about 170 years ago, its surface elevation is still increasing, contributing to positive mass balance. In this study, we considered the potential of the Totten and Vanderford glaciers, East Antarctica, to undergo such a reconfiguration. We find that the drainage divide between the Totten and Vanderford glaciers is transient and that relatively minor changes in the ice sheet geometry could cause ice flow piracy from Totten to Vanderford Glacier. Our assessment of the basal water routing suggests that basal water piracy could also occur under larger ice sheet geometry changes, although this estimate is preliminary. The longer-term impacts of potential piracy from Totten to Vanderford Glacier on the flow dynamics, and hence overall mass balance, of this region is unknown. Given our finding that ice surface geometry changes can drive migration of the Totten-Vincennes Bay catchment boundary, determining how fast such changes might occur and their subsequent impacts on ice mass loss from the ASB and consequent global sea level rise deserves further investigation. If ice flow and basal water piracy is a result of coupled ice sheet-subglacial hydrology interactions, then without inclusion of these processes we may not be able to predict the occurrence of piracy here or elsewhere in Antarctica. The findings motivate further investigation into other regions of Antarctica that may be vulnerable to substantial flow reconfiguration, including the relevant processes that drive these changes and the timescales over which they could occur.



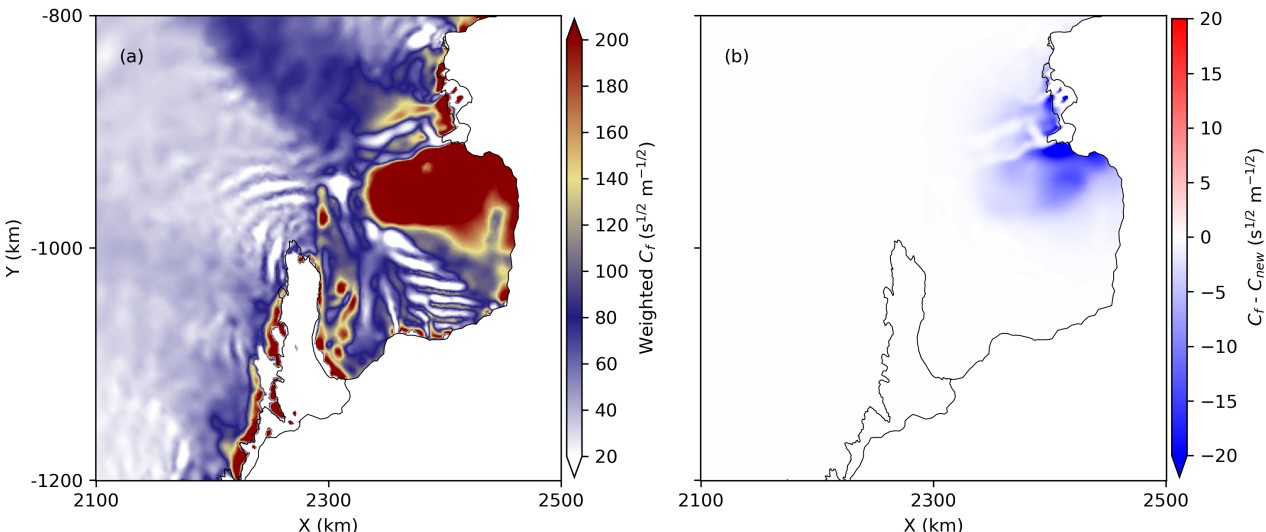

**Figure A1.** (a) Basal friction coefficient $C_f$ ($\mathrm{s}^{1/2}\,\mathrm{m}^{-1/2}$) calculated using inversion for the spin-up and weighted by $M$; (b) difference between $C_f$ and $C_{new}$ for $\sigma = 0.1$ ($\mathrm{s}^{1/2}\,\mathrm{m}^{-1/2}$).

*Data availability.* All of the datasets used in this study are publicly available. We use version 4.20 of the open source ISSM software, which is freely available for download from https://issm.jpl.nasa.gov/. The datasets used to initialise the model are available via the corresponding articles cited in this paper.

## Appendix A: Friction coefficient reduction

450 In Eqn. 4 for the perturbed friction coefficient field, we choose $\alpha = 20$ to concentrate the reduction in basal friction close to the Vanderford Glacier grounding line, as it corresponds well with the region of maximal, weighted friction, $MC_f$ (i.e. the region where $MC_f > 100\,\mathrm{s}^{1/2}\,\mathrm{m}^{-1/2}$). We note that this impacts the depth of penetration of changes in the surface elevation into the ASB, as discussed in Sect. 5.



## Appendix B: The effective pressure calculation

We use the parameterisation for the effective pressure ($N_E$) defined in McArthur et al. (2023), which is given by:

$$N_E = \rho_{ice} g h (1 - r_l) \frac{\tilde{h}^m}{\tilde{h}^m + h^m}, \tag{B1}$$

$$\tilde{h} = \left( \frac{1 - \gamma}{\gamma - r_l} \right)^{1/m} h_t, \tag{B2}$$

$$m = \frac{\ln\left(\frac{1 - r_l}{\epsilon} - 1\right) + \ln(\gamma - r_l) - \ln(1 - \gamma)}{\ln(h_t) - \ln(h_s)}. \tag{B3}$$

Here, $\rho_{ice}$ is the density of ice (kg m$^{-3}$), $g$ is the gravitational acceleration (m s$^{-2}$) and $h$ is ice thickness (m). Both $\tilde{h}$ and $m$ are constants; $\gamma$ is a constant representing the typical effective pressure as a fraction of the ice overburden pressure for regions of thick ice; $r_l$ is the basal water pressure as a fraction of the ice overburden pressure for ice thicknesses tending to zero; $h_t$ is a typical thickness value for thick ice; $h_s$ is a typical thickness value for thin ice; and $\epsilon$ is a small constant chosen so that the water pressure in regions of thin ice is low $(r_l + \epsilon)\rho_{ice} g h$. To find values for the constants $r_l$, $\gamma$, $h_s$, and $h_t$, and for $\epsilon = 0.05$, we use output from the GlaDS model from Dow et al. (2020) for the Totten and Vincennes Bay catchments, and similarly to McArthur et al. (2023), this results in $r_l = 0.7$, $\gamma = 0.96$, $h_s = 500$ m, and $h_t = 2800$ m.

*Author contributions.* FSM conceived and led the study, conducted the ice sheet modelling and analysis, and prepared the manuscript. BK contributed to manuscript planning. JLR assisted with ice sheet modelling methodology. KM provided analysis on the new effective pressures. All authors contributed to the preparation and editing of the manuscript.

*Competing interests.* At least one of the (co-)authors is a member of the editorial board of The Cryosphere.

*Acknowledgements.* We thank Poul Christoffersen for helpful discussions that improved the manuscript. FSM was supported under an Australian Research Council (ARC) Discovery Early Career Research Award (DECRA; DE210101433); RSJ was supported under a DECRA (DE210101923). FSM, AM, RSJ and LB were supported under the ARC Special Research Initiative (SRI) Securing Antarctica's Environmental Future (SR200100005). This research was undertaken using resources from the National Computational Infrastructure Merit Allocation Scheme, supported by the Australian Government. BK, AA and KH were supported by the ARC SRI, Australian Centre for Excellence in Antarctic Science (SR200100008). CFD was supported by the Natural Sciences and Engineering Research Council of Canada (NSERC RGPIN- 03761-2017) and the Canada Research Chairs Program (CRC 950-231237).




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
