# Peer review of "Assessing the potential for ice flow piracy between Totten and Vanderford glaciers, East Antarctica"

_EGUsphere, 2023_

## Author Comment (AC1)

**Authors response to reviews on *Assessing the potential for ice flow piracy between Totten and Vanderford glaciers, East Antarctica**

*Felicity S. McCormack, Jason L. Roberts, Bernd Kulessa, Alan Aitken, Christine F. Dow, Lawrence Bird, Ben K. Galton-Fenzi, Katharina Hochmuth, Richard S. Jones, Andrew N. Mackintosh, and Koi McArthur*

We are glad that both reviewers found the manuscript so easy to read and clear and we thank both reviewers for their comments and suggestions.

In what follows, the reviewer comments are in black and our response, including proposed changes to the manuscript, are in blue.

Sincerely,

Felicity McCormack and co-authors.

**Response to Reviewer 1**

Review of "Assessing the potential for ice flow piracy between Totten and Vanderford glaciers, East Antarctica". This paper uses an ice sheet model (ISSM) to study the impact of changing surface elevation on ice and water flow between the Totten and Vanderford glaciers. The changes in surface elevation are not model predictions, but rather reasonable forcing parameters, allowing the impact of such changes to be assessed. The authors find that a proportion of ice and water flow can be switched from Totten to Vanderford Glacier, with the most significant changes driven by increasing the elevation within the Totten Glacier catchment. The authors indicate that these findings are significant as they show that water and ice piracy are possible within East Antarctica. The impact of such piracy on ice flow was not considered by this study, but is put forward as an important factor to be included in future models of the region.

I found the paper clear and well written and that the conclusions are supported by the data and model outputs.

I have a number of minor comments/suggestions which I feel may help the reader to better understand the paper.

L20-21 states "which contains approximately 7m of global sea level equivalent (Morlighem et al., 2020) of which 3.5m is grounded below sea level". This is a little confusing, what people want to know is what is the actual sea level rise they could

experience - maybe split out the end of the sentence to say, "As much of the catchment lies below sea level, the effective impact of total deglaciation of this region would be at least 3.5m of global sea level rise".

We propose to modify the sentence as follows:

"... which contains approximately 7 m of global sea level equivalent (Morlighem et al., 2020). **However, half of that water-equivalent ice volume lies on topography that is grounded below sea level, and hence the ASB is highly vulnerable to significant deglaciation as the climate warms.**"

L24 states "Paleoclimate evidence suggests markedly reduced ice volumes in the ASB…." Being picky - do you mean "Ice sheet models driven by realistic temperatures derived from paleoclimate evidence.....". Not sure that Paleoclimate evidence can directly suggest reduced ice volumes. Paleo sea-level records would suggest reduced ice volumes, but are hard to attribute to specific basins. Paleoclimate observations close to the ASB/WSB ect showing elevated temperatures could be considered more direct paleoclimate evidence for ice free areas, if so this could be stated.

Thanks for raising this. It is both geological and ice sheet modelling evidence, and the geological data come from studies that look at sediment records proximal to the East Antarctic coastline and the provenance of these sediments. We removed reference to the Recovery Basin as it's outside the domain of interest for our study. We suggest the following edits to be more clear (from L24):

"**Geological data proximal to the East Antarctic coastline from Princess Elizabeth Land to Adélie Land (Cook et al., 2013, Williams et al., 2010, Wilson et al., 2018)** suggests markedly reduced ice volumes could have been present in the ASB and Wilkes Subglacial Basin during the mid-Pliocene warm period (MPWP; 3.3 Ma to 3 Ma), when global temperatures were 2.5 to 4°C warmer than the 1850 to 1900 mean. **Ice sheet model simulations of the MPWP show preferential ice loss from the ASB and Wilkes Subglacial Basins (Austermann et al., 2015, DeConto and Pollard, 2016),** highlighting the potential of the ASB to reach a tipping point as the climate warms (McKay et al., 2022, Noble et al., 2020)."

L37 to L46 I would flag that the work referred to in this paragraph was carried out on the Siple Coast. Later in the paragraph when the authors state "modelling of this system" they could say "modelling of the Siple Coast system". This would avoid confusion with "this system" being the current study area.

Excellent suggestion. We propose to amend as follows:

"Changes in basal water accumulation and routing have been hypothesised to play a role in flow diversion (piracy) – and even ice flow stagnation – between neighbouring ice streams **of the Siple Coast** in the past."

and

"More recent modelling **of the Siple Coast** system…"

Fig. 1. Sabrina Subglacial Basin (SSB); Sabrina Coast (SC); Knox Coast (KC); and the Elcheikh saddle point (black X) are mentioned in the caption, but not shown on any figure. These are important as they are directly mentioned in the text later, e.g. L78.

We will update Figure 1 to include these locations.

Fig. 9 It could be worth showing the ice catchment boundaries from Figs 1 & 2 on these maps for context. Also it might be worth outlining the key hydrological catchments to highlight the significant switch in the area drained.

Great suggestion. We will add ice catchment boundaries to figure 9, and to draw the hydrological catchments in a different colour.

L435-436 States "We find that the drainage divide between the Totten and Vanderford glaciers is transient and that relatively minor changes in the ice sheet geometry could cause ice flow piracy from Totten to Vanderford Glacier". It would be nice to say here how far the catchment boundary was shifted for context.

We propose the following change:

"We find that the drainage divide between the Totten and Vanderford glaciers is transient and that relatively minor changes in the ice sheet geometry could cause ice flow piracy from Totten to Vanderford Glacier. **For example, increasing the SMB at Totten Glacier for scenarios of between 1 and 7°C of warming (SMB experiments), for corresponding magnitudes of thinning at the Vanderford Glacier (friction experiments), and for combinations of the two effects (combined experiments), leads to an increase in the Vincennes Bay catchment area of between 3 and 22%. The catchment boundary shifts up to 275,449 km for the SMB experiments, up to 30,019 km for the friction experiments and up to 284,165 km for the combined experiments, where the maximum boundary migration occurs at the most inland point the catchment and the minimum distance at the Elcheikh saddle point."**

Figure A1. Does not appear to be referenced in the text.

We will add citation of Figure A1.

**Response to Reviewer 2**

Review of 'Assessing the potential for ice flow piracy between Totten and Vanderford glaciers, East Antarctica'

This is a very well written manuscript that highlights the potential for ice flow reconfiguration for a paor of glaciers in East Antarctica, a phenomenon that has happened for glaciers on Siple Coast.

The authors summarize the state of knowledge needed to motivate and support the work undertaken in this study. The project is well described, results are discussed in sufficient detail and provide a clear set of conclusions.

Model based as well as remote sensing based inputs are used, they all are discussed and evaluated for strengths, shortcomings, and needed improvements. Equally important to the actual findings of the study is the discussion of the information needs for the region:

"Improved knowledge of the bed geometry and sediment characteristics in these regions is essential, and should be a focus of future airborne surveys"

Potential improvements for SMB estimates

"Given that accurate knowledge of ocean melt rates is critical for ice sheet model simulations of retreat, and the evidence for a strong observed warming trend of mCDW in this region of East Antarctica "… ", this emphasises the need for long-term ocean state monitoring in this region."

I highlight these because rather than bemoaning the quality of available products, which are the current state of knowledge, the authors make due with what is available, and discuss the impact of accuracies and errors in the input data. In addition, the authors highlight needed improvements and challenge the community as well as funding agencies with a very clear set of information needs from both regional climate models and remote sensing campaigns.

Figure 1: Areas mentioned in the caption are not shown in the figures  (SBB, SC, the X for Elcheikh Saddle Point)

Thanks for picking this up. We will add these locations to figure 1.

You mention Law Dome multiple times in the text, this should accordingly also be indicated in one of the figures.

We will notate Law Dome on figure 1, and again on figure 6 (to correspond with the text in section 3.2).

Highland C: The "C" is not well legible

We will move the label to make sure it's legible.

Figure 8: Are catchment divide changes only happening between Totten and Vanderford glaciers? Figure 9 indicates very few changes in the water routing between Totten and Moscow University glaciers, but did you look at divide changes there?

This is a great point. Our sensitivity analyses did not consider the effect of changes in the Moscow catchment, so we are unable to make a sound/justified conclusion on the potential for flow piracy between the Totten and Moscow University Ice Shelf (MUIS) glaciers with our current results. We pursued our experimental design on the sediment records offshore of the Sabrina and Knox Coasts, as these sediment records are able to clearly distinguish between

contributions from the Totten and Vanderford Glaciers. We are unfortunately unable to distinguish between the contributions of the Totten and MUIS glaciers to the Sabrina Coast sediment records, although this motivates the collection of sediment cores on the continental shelf proximal to these two glaciers, from which we might be able to obtain a higher resolution, more precise estimate of sedimentation.

Given that most of the current ice mass loss in the Aurora Subglacial Basin is concentrated in the Totten and MUIS glaciers (e.g. Smith et al., 2020), and that the Moscow catchment contains about one third of the Totten Catchment (approximately twice that of the Vincennes Bay catchment), it would be very interesting to explore the potential for flow piracy between the Totten and MUIS glaciers in future studies, e.g. in the context of a transient forced simulation of a coupled ice sheet-subglacial hydrology model.

---

## Author Response (AR1)

1 September 2023

**Manuscript updates to *Assessing the potential for ice flow piracy between Totten and Vanderford glaciers, East Antarctica**

*Felicity S. McCormack, Jason L. Roberts, Bernd Kulessa, Alan Aitken, Christine F. Dow, Lawrence Bird, Ben K. Galton-Fenzi, Katharina Hochmuth, Richard S. Jones, Andrew N. Mackintosh, and Koi McArthur*

Dear Dr MacGregor,

Please find attached the updated manuscript, with revisions as per our response document on 21 August 2023.

In the attached updated manuscript we have also edited figure 3 – the colour scale to make it easier for visualisation, and the legend to make it more clear. A number of other minor edits were also made (e.g. correcting unit typos in table 1).

Sincerely,

Felicity McCormack and co-authors.